# Ranking vs. Classifying:
# Measuring Knowledge Base Completion Quality

**Marina Speranskaya**                                    SPERANSKAYA@CIS.LMU.DE
**Martin Schmitt**                                               MARTIN@CIS.LMU.DE
**Benjamin Roth**                                               BEROTH@CIS.LMU.DE
*Center for Information and Language Processing, LMU Munich, Germany*

## Abstract

Knowledge base completion (KBC) methods aim at inferring missing facts from the information present in a knowledge base (KB). Such a method thus needs to estimate the likelihood of candidate facts and ultimately to distinguish between true facts and false ones to avoid compromising the KB with untrue information. In the prevailing evaluation paradigm, however, models do not actually decide whether a new fact should be accepted or not but are solely judged on the position of true facts in a likelihood ranking with other candidates. We argue that consideration of binary predictions is essential to reflect the actual KBC quality, and propose a novel evaluation paradigm, designed to provide more transparent model selection criteria for a realistic scenario. We construct the data set FB14k-QAQ with an alternative evaluation data structure: instead of single facts, we use KB queries, i.e., facts where one entity is replaced with a variable, and construct corresponding sets of entities that are correct answers. We randomly remove some of these correct answers from the data set, simulating the realistic scenario of real-world entities missing from a KB. This way, we can explicitly measure a model's ability to handle queries that have more correct answers in the real world than in the KB, including the special case of queries without any valid answer. The latter especially contrasts the ranking setting. We evaluate a number of state-of-the-art KB embeddings models on our new benchmark. The differences in relative performance between ranking-based and classification-based evaluation that we observe in our experiments confirm our hypothesis that good performance on the ranking task does not necessarily translate to good performance on the actual completion task. Our results motivate future work on KB embedding models with better prediction separability and, as a first step in that direction, we propose a simple variant of TransE that encourages thresholding and achieves a significant improvement in classification $F_1$ score relative to the original TransE.

## 1 Introduction

A knowledge base contains relational information about the world in the form of triples. For instance, the fact "New York is located in the USA" could be represented as the triple (`New_York`, `located_in`, `USA`). Given the available information in an incomplete knowledge base, the task of knowledge base completion (KBC) is to find missing facts by predicting the most likely missing relation between known entities. Formally, a knowledge base describes a set of objects - or *entities* - $\mathcal{E}$, connected to each other via binary *relations* $\mathcal{R}$, and contains a collection of supposedly true facts $KB^+ \subseteq \mathcal{E} \times \mathcal{R} \times \mathcal{E}$. The KBC task is to infer new true facts consisting of *head entity* $h'$, *relation* $r'$ and *tail entity* $t'$ with $(h', r', t') \notin KB^+$,

given the facts in $KB^+$. The quality of KBC is typically measured by removing a triple $(h, r, t)$ from $KB^+$ and comparing the score assigned by a completion algorithm to the scores assigned to perturbed triples $(h, r, t')$ where $t' \neq t$.

Evaluation of embedding models on the KBC task intuitively should measure the quality of facts added by a completion algorithm. Standard metrics for evaluating KBC such as top-k precision or mean reciprocal rank (MRR), however, measure the quality of *ranking* possible knowledge graph triples. These metrics do not necessarily reflect the real performance of the underlying task since it would be necessary to combine a triple scoring mechanism (e.g., based on knowledge graph embeddings) with thresholds for obtaining a prediction mechanism, and the triple scoring mechanism might not be *consistently scaled*: It could be the case that the relative ranking of tuples may be satisfactory, when ranked for the same query tuple (h, r, ?), but that scores are not well-calibrated and finding good global or per-relation thresholds is difficult for certain embedding-based scoring mechanisms.

In this work, we propose an alternative way of evaluating KBC quality by reporting classification measures (e.g., $F_1$ scores) on the carefully constructed data set FB14k-QAQ that balances query tuples for which completion is possible with special query tuples that, by construction (using type constraints and entity removal), are impossible to complete. This new evaluation approach motivates research on embedding models that intrinsically support thresholds for prediction and we propose a simple variant of TransE that improves on the new evaluation metric relative to the original model.

Previous work [Socher et al., 2013] has attempted to overcome problems of ranking-based evaluation approaches by artificially creating a fixed amount of negative samples from positive triples (typically a 1-1 ratio) and measuring accuracy on that data set. Such a setting, however, does not properly reward models that are able to distinguish between relationships that should have more positive predictions vs. those that should have less. Wang et al. [2019] identify problems of previous evaluation paradigms based on entity rankings and they propose an alternative scheme that looks at all possible entity pairs, ranked for a given relation. While their proposed evaluation solves some of the problems (comparabilty of scores between query entities), it is still ranking-based and does not incentivize the scoring model to support globally optimal prediction thresholds across relationships.

The main contributions of this paper are:

- We construct a data set for extensive classification evaluation that penalizes models that predict erroneous triples. For this, we construct two types of *negative* cases: First, a subset of entities is sampled and removed from an existing knowledge base, so that corresponding queries can be obtained that, by construction, do not have any correct answers. Second, we formulate queries that violate type constraints and thus cannot have any right answers either.

- Experiments with established embedding models show surprising differences when the new metric is compared to an evaluation based on ranking.

- Our evaluation suggests that models should focus on optimizing separability of their predictions. An adapted version of TransE that encourages separability improves by over 30% $F_1$ score relative to the original model.

## 2 Related work

**Knowledge graph embedding models.** Embedding models assign a latent representation to every entity and relation of a knowledge base. Within the scope of this paper, entities $h \in \mathcal{E}$ are represented as $d$-dimensional vectors $\mathbf{e_h} \in \mathbb{R}^d$ and relations $r \in \mathcal{R}$ either as vectors $\mathbf{r_r} \in \mathbb{R}^d$ or as matrices $\mathbf{R_r} \in \mathbb{R}^{d \times d}$. A KBC model is characterized by its scoring function $s(h, r, t) : \mathcal{E} \times \mathcal{R} \times \mathcal{E} \to \mathbb{R}$.

Various approaches exploiting such representations of entities and relations have been proposed for the task of knowledge base completion. One of the most prominent group among them is that of *tensor factorization models*: RESCAL [Nickel et al., 2011] with the scoring function $s(h, r, t) = \mathbf{e_h}^T \mathbf{R_r} \mathbf{e_t}$, DistMult [Yang et al., 2014], which sets diagonal restrictions to the matrix, with $s(h, r, t) = \mathbf{e_h}^T diag(\mathbf{r_r}) \mathbf{e_t}$, and ComplEx [Trouillon et al., 2016], which uses complex-numbered embeddings with the previous scoring function. The field of *translation models* was opened by the TransE [Bordes et al., 2013] scoring approach $s(h, r, t) = -\|\mathbf{e_h} + \mathbf{r_r} - \mathbf{e_t}\|_p$, followed by a number of variants, such as projection on relation-specific hyperplanes (TransH [Wang et al., 2014]) or transforming entity embeddings to a relation-specific vector space prior to scoring (TransR [Lin et al., 2015]). TransA [Xiao et al., 2015] scoring uses an additional matrix $\mathbf{W_r}$ per relation, which is derived from the entity and relation embeddings analytically (rather than learned), and also replaces the $L_p$-norm: $s(h, r, t) = -(|\mathbf{e_h} + \mathbf{r_r} - \mathbf{e_t}|)^T W_r(|\mathbf{e_h} + \mathbf{r_r} - \mathbf{e_t}|)$, where $|\mathbf{e_h} + \mathbf{r_r} - \mathbf{e_t}|$ takes an absolute value in every vector position. More recently, further improvements were obtained with neural approaches like ConvE [Dettmers et al., 2017], KG-Bert [Yao et al., 2019] and Graph Attention Networks [Nathani et al., 2019]. For the purposes of this work, we selected a cross-group model sample consisting of DistMult, ComplEx, TransE and ConvE.

**Alternatives to ranking-based prediction.** Socher et al. [2013] first introduced an evaluation setting based on triple classification. However, the data contains only two randomly sampled negative triples for every positive triple, which poses an unrealistic ratio for KBC. Additionally, in case of overlapping negative samples $(h, r, t')$ for two test triples $(h, r, t_1)$ and $(h, r, t_2)$, the evaluation protocol would count the predicted labels of these negative samples twice into the final classification metric. This redundancy effect would be even stronger with a higher number of negatives samples due to a higher overlap probability. Our query-based evaluation, however, eliminates it completely as all $(h, r, ?)$ test triples are considered at once.

In contrast to KB embedding methods, Godin et al. [2019] approach query answering with a reinforcement learning model that attempts to find a path through a knowledge graph from $h$ to the correct $t$. Their approach allows the model to refrain from giving an answer rather than giving a wrong one. For evaluation, they suggest to use the precision of the given answer together with an answer rate metric that measures the rate of empty responses of the model. However, the used data set contains exactly one correct answer for each given query, missing other realistic cases, such as multiple answers and unanswerable queries, for which these metrics would not be applicable.

**MRR reviews.** Sun et al. [2019] have noticed inconsistencies in model behavior measured by mean reciprocal rank (MRR) that the authors attribute to the ranking setting itself. An inappropriate performance measure can become an explanation for the sudden come-

backs of rather basic models in Kadlec et al. [2017] and Ruffinelli et al. [2020], as well as strong performance variations. In their recent work, Wang et al. [2019] criticize the current evaluation protocol with the mean reciprocal rank for being unsuitable for KBC. They accuse it of overestimating the model performance due to its insensitivity to unrealistic and nonsensical triples and they propose an improved — but still ranking based — metric for KB embedding evaluation.

## 3 Data set FB14k-QAQ

In this section a new evaluation setting is proposed that directly measures the quality of extracted facts by matching against a ground truth. For this, care must be taken that evaluation not only accounts for correctly retrieved facts, i.e., true positives, but also rewards cases where the model correctly refrains from predicting an answer, i.e., produces less false positives. We argue that a query-driven setup with carefully constructed query and answer sets is necessary for measuring KBC quality on the triple level. As a foundation for the FB14k-QAQ data set (measuring **Q**uery **A**nswering **Q**uality), we base our work on the FB15k-237 [Toutanova et al., 2015] data set, a subset of FreeBase, because it is a well-established benchmark for KBC and has been used in most of the recent publications on KB embedding methods.

**From facts to queries.** The new evaluation setting relies on *queries*. A query is an entity-relation pair where a second entity is missing to form a complete triple: if the tail position is open for completion, we call such a query $q_1 = (h, r, ?)$ an *tail query*. A query of the form $q_2 = (?, r, t)$ is called *head query*, respectively. The ∘ operator is defined to fill the open position of a query with the specified entity $i \in \mathcal{E}$, i.e., $q_1 \circ i = (h, r, i)$ and $q_2 \circ i = (i, r, t)$. For every query $q$, the set of correct answers $\mathcal{A}_q^F$ can be defined given a set of valid facts $F \subseteq \mathcal{E} \times \mathcal{R} \times \mathcal{E}$ by setting $\mathcal{A}_q^F := \{i \mid (q \circ i) \in F\}$.

Figure 1 depicts the data set creation process, which operates on the original training data and the data from the original development and test splits (further, evaluation data): (1) A small subset of entities $\mathcal{E}^- \subset \mathcal{E}$ from the original data set is selected for removal ("Select"). We will refer to the remaining entities as $\mathcal{E}^+ := \mathcal{E} \setminus \mathcal{E}^-$. (2) In "Split", the new training set is obtained by taking all original training triples that only contain entities from $\mathcal{E}^+$. The evaluation data and the remaining training triples are used to create queries and determine answer sets, i.e., they form the set of valid facts $F$ mentioned above. (3) The facts from $F$ are then grouped by head and relation to form tail queries and by tail and relation for head queries, respectively. The "Group" step results in a set of queries $q$ with corresponding answer sets $A'_q$. (4) To obtain the final answer sets $A_q$, the selected entities $\mathcal{E}^-$ are removed from the answer sets in "Remove". With respect to the intermediate answer sets $A'_q$ from "Group" and the final ones $A_q$ from "Remove", the queries can be divided into two sets: $\mathcal{C}$, where the answer set remained complete, and $\mathcal{I}$, where entities have been removed from the answer sets (including the special case of empty answer sets $\mathcal{N} \subset \mathcal{I}$). The full formal description of the process is provided in Appendix A.1.

By removing entities from real triples of the original data set, we artificially create a situation where completion entities exist in the real world but are not present in the data set, thereby simulating a controlled closed-world problem. It constitutes a challenge to KBC

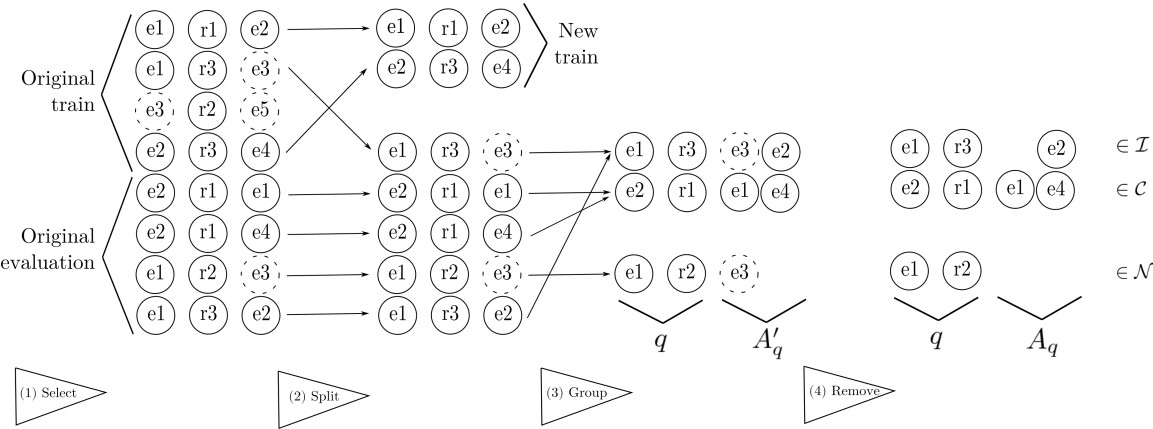

Figure 1: Data set transformation scheme as described in Section 3 "From facts to queries".
Step (3) only shows the grouping for head queries; tail queries are constructed
analogously. Full circles represent relations and entities from $\mathcal{E}^+$; entities from $\mathcal{E}^-$,
i.e., those selected for removal, are dashed ($e3$ and $e5$). Evaluation data is a union
of the original development and test data. Last column shows set membership: $\mathcal{C}$
if $A'_q = A_q$, $\mathcal{I}$ otherwise (edge case $\mathcal{N} \subset \mathcal{I}$ if $|A_q| = 0$).

models not to complete such a query that not only appears meaningful but actually has a
real-world answer to it outside of the knowledge base.

**Queries with type violation.** A more relaxed version of queries with empty answer
sets are queries with an inherent contradiction that would immediately be recognized
by a human, such as (Albert_Einstein, has_capital, ?). The formal contradiction in
this query is expressed in the type system of FreeBase triples. In FreeBase, entities are
labelled with different *types*, e.g., an entity $h = $ New_York_City is assigned a type set
$\mathcal{T}_h = \{\text{LOCATION, ART\_SUBJECT, WINE\_REGION}\}$ while for $t = $ Albert_Einstein it is $\mathcal{T}_t = $
$\{\text{PERSON, BOOK\_AUTHOR, SCIENTIST}\}$. The type system ensures that every relation only
takes entities of a specific type as head and tail argument, respectively. For instance, a
relation $r = $ has_capital takes entities of type $dom(r) = $ COUNTRY for its head position
(relation domain) and entities of type $rng(k) = $ CITYTOWN for its tail position (relation
range). Triples in FreeBase follow this scheme and are therefore type-consistent, i.e.,
$KB^+ \subseteq \{(h, r, t) \mid dom(r) \in \mathcal{T}_h, rng(r) \in \mathcal{T}_t\}$. However, a FreeBase triple can be type-
consistent and still false, e.g., (USA, has_capital, New_York_City).

By analogy, type-consistent queries obey domain and range restrictions with respect to
the already filled position and type-inconsistent queries do not. An obviously incorrect query
(Albert_Einstein, has_capital, ?) in fact violates the types since the domain COUNTRY of
has_capital does not occur in the type set $\mathcal{T}_t$ for the entity $t = $ Albert_Einstein. Formally,
a set of such type-inconsistent "fake" queries $\mathcal{F}$ can be characterized as follows:

$$\mathcal{F} \subset \{(h, r, ?) \mid h \in \mathcal{E}^+ \wedge dom(r) \notin \mathcal{T}_h\} \cup$$
$$\{(?, r, t) \mid t \in \mathcal{E}^+ \wedge rng(r) \notin \mathcal{T}_t\}$$

| Query set | Head | Tail | Total |
|---|---|---|---|
| $\mathcal{C}$ | 8,778 | 15,688 | 24,466 |
| $\mathcal{I}$ | 10,676 | 13,144 | 23,820 |
| $\mathcal{N}(\subset \mathcal{I})$ | 6,871 | 8,850 | 15,721 |
| $\mathcal{F}$ | 5,712 | 11,132 | 16,844 |

Table 1: Data set queries breakdown by the type of the query answer set ($\mathcal{C}$ - complete answer set, $\mathcal{I}$ - entities have been removed from the answer sets, $\mathcal{F}$ - queries with type violations).

This means, all queries violate at least one of the relation domain or range restrictions[1] and thus cannot have a valid completion due to this type inconsistency. In case of an automated knowledge base completion setting, e.g., when there is no type information available to check completion queries for type violations, this can become quite relevant. A perfect model should distinguish well-typed queries from nonsensical ones. By combining type-consistent $\mathcal{N}$ and type-violating $\mathcal{F}$ queries with empty answer sets, we address different aspects of model behavior.

**Overview.** The number of entities in $\mathcal{E}^-$ is an essential parameter within this data set construction strategy. The more entities are removed, the more queries with smaller answer sets potentially arise. We pursued a specific final distribution of queries: queries with no answer $\mathcal{N}$ make 25% of the evaluation set, queries with at least one answer make 50%; the remaining 25% are filled with type-violating queries $\mathcal{F}$. For the FB14k-QAQ data set, we achieved this by removing $|\mathcal{E}^-| = 1000$ entities.

The resulting FB14k-QAQ data set has 13,541 entities and 237 relations, with 236,795 triples in the training set.[2] Three sets of different query groups $\mathcal{C}$, $\mathcal{I}$, and $\mathcal{F}$ are evenly split between the development $\mathcal{D}$ and the test set $\mathcal{T}$, resulting in 32.5k queries each, 50 percent of which have an empty answer set. The other 50 percent include queries with one or more correct answers. The ratio of queries with a complete answer set $\mathcal{C}$ to those with a removed answer $\mathcal{I}$ is almost 1:1; more detailed statistics about the query distribution between sets are presented in Table 1.

## 4 Evaluation

**Thresholding.** To measure the models' capability to decide between true and false triples, a threshold $\tau_r$ is applied to the output scores to obtain binary predictions. We consider two thresholding settings: (i) The same $\tau_r = \tau_{global}$ value is shared across all relations and (ii) $\tau_r$ is relation-specific. The global threshold can be easily optimized for a given set of predictions. The relation-specific thresholds are found using a greedy iterative algorithm (details in Appendix B.1) that optimizes the micro-averaged $F_1$ score for 474 relations (including the separately embedded inverse relations).

---

1. Type information provided by [Wang et al., 2019].
2. The final data set, as well as the source code for data set construction and model evaluation are available at https://github.com/marina-sp/classification_lp.

**MRR.** The query-based format of the FB14k-QAQ is incompatible with the MRR evaluation. To be able to evaluate this data set on ranking, we reconstruct the underlying valid facts $F$ from the queries in dev $\mathcal{D}$ and test $\mathcal{T}$ data by completing every query with all entities from their answer set, resulting in triple sets

$$\mathcal{D}_{rank} = \{q \circ i \mid q \in \mathcal{D} \wedge i \in A_q\}$$

and $\mathcal{T}_{rank}$ analogously. The empty queries $\mathcal{N}$ and $\mathcal{F}$ are ignored in the ranking evaluation, as they cannot build a valid fact.

The triples *(h, r, t)* from $\mathcal{D}_{rank}$ and $\mathcal{T}_{rank}$ are scored and ranked against all possible perturbations of entities as follows: The rank $rank(h,r,t)$ of an evaluation triple *(h, r, t)* is defined as its index in a sorted array of scores. For perturbed tails, this is $\{(h,r,t') \notin Train \mid t' \in \mathcal{E}^+\}$; for heads, it is $\{(h',r,t) \notin Train \mid h' \in \mathcal{E}^+\}$, i.e., scores for known triples from the training set are excluded. The mean reciprocal rank for an evaluated set of triples $X_{rank}$ (substitute $D_{rank}$ and $T_{rank}$) is then

$$MRR_E = \frac{1}{|E_{rank}|} \sum_{(h,r,t) \in E_{rank}} \frac{1}{rank(h,r,t)}$$

**Metric definition.** In the classification setting, evaluation is based on a model's binary decisions. For an arbitrary query $q$ with a relation $r$, all entities with a score $s$ above the tuned threshold $\tau_r$ constitute the positive response set

$$R_q = \{i \in \mathcal{E}^+ \mid s(q \circ i) > \tau_r\},$$

i.e., the model retrieved these entities as a valid query completion.

Recall from Section 3 that, for each query $q$, the evaluation data contain a set $A_q$ of expected correct (relevant) answers that were not directly seen in the training data. In order not to punish a model for correctly reproducing the facts from the train data, the corresponding entities are excluded from the retrieved set:

$$R_q \leftarrow R_q \setminus \{i \in \mathcal{E}^+ \mid (q \circ i) \in Train\}$$

To assess the retrieval quality of the model on a set of queries $\mathcal{X}$ (substitute $\mathcal{D}$ or $\mathcal{T}$), we define the following sets that describe the correctly retrieved entities (true positives), erroneously retrieved entities (false positives), and entities missing in the retrieved set (false negatives) for a query $q$:

$$TP_q = |R_q \cap A_q| \qquad FP_q = |R_q \setminus A_q| \qquad FN_q = |A_q \setminus R_q|$$

With $TP = \sum_{q \in X} TP_q$, $FP = \sum_{q \in X} FP_q$ and $FN = \sum_{q \in X} FP_q$ the micro-averaged precision, recall and $F_1$ score can be easily computed. We use $F_1$ score as the final performance measure.

## 5 Results

**Experimental setup.** The framework for this work was built on top of the publicly available ConvE implementation.[3] We used the provided implementations for ConvE,

---

3. https://github.com/TimDettmers/ConvE

| Model ($d$) | MRR | $F_1$ global threshold | | | | $F_1$ multiple thresholds | | | |
|---|---|---|---|---|---|---|---|---|---|
| | | full | $\mathcal{C}$ | $\mathcal{C} \cup \mathcal{F}$ | $\mathcal{I}$ | full | $\mathcal{C}$ | $\mathcal{C} \cup \mathcal{F}$ | $\mathcal{I}$ |
| ConvE 128 | **.321** | .134 | .272 | .211 | .105 | **.204** | **.317** | **.286** | .150 |
| ConvE 64 | .263 | .157 | **.307** | .261 | .108 | .189 | .312 | .280 | .135 |
| ComplEx 128 | .293 | .021 | .169 | .017 | .042 | .190 | .296 | .261 | .143 |
| ComplEx 64 | .293 | .009 | .157 | .005 | .057 | .181 | .282 | .245 | .143 |
| TransE 128 | .293 | .108 | .158 | .106 | .108 | .159 | .172 | .168 | **.154** |
| TransE 64 | .283 | .111 | .164 | .112 | **.110** | .161 | .176 | .175 | **.154** |
| DistMult 64 | .266 | **.159** | .273 | **.226** | .129 | .184 | .256 | .239 | .148 |
| DistMult 128 | .221 | .133 | .275 | .188 | .088 | .163 | .206 | .194 | .138 |

Table 2: MRR and $F_1$ score on the full FB14k-QAQ test set together with $F_1$ scores on different query subsets in the two threshold settings. "$F_1$ global threshold" refers to scores obtained with a single shared threshold for all relations while "$F_1$ multiple thresholds" refers to the setting with independent thresholds for every relation (including inverse relations).

ComplEx and DistMult. Similarly to ComplEx and DistMult, we provided the traditional TransE with an additional top layer, which transforms real-numbered scores to a probability-like output.[4]

All models share the following settings: Adam optimizer, binary cross entropy loss, *KvsAll* training[5], and a maximum number of 200 training epochs. Development loss is used as the early stopping criterion with patience of 50 epochs.[6] The other hyperparameters were selected from the following value sets: embedding dimension $d$ from $\{64, 128\}$[7], batch size from $\{256, 512, 1024\}$, learning rate from $\{0.001, 0.0001\}$, inverse relations from $\{$yes, no$\}$ (for TransE only). Entity embedding L2-normalization and L1-scoring was used for TransE as in the original model. Optimal threshold values are provided in Appendix B.2.

**Discussion.** Table 2 shows the evaluation results of standard embedding models on the test data according to the classic ranking metric MRR and according to the $F_1$ score as proposed above. Development results are provided in Appendix C.2. It is evident that MRR and the suggested classification-based evaluation scheme assess the evaluated model variants in a strikingly different manner.

A good ranking as measured by MRR is not necessarily a good indicator for a good performance in the KBC settings: The ComplEx models have a relatively high MRR but show the weakest performance in the globally thresholded prediction ($F_1$ global threshold

---

4. ComplEx and DistMult have a sigmoid prediction layer. Since the TransE distances are non-negative, a hyperbolic tangent function is more appropriate choice than a sigmoid and exploits the whole interval [0,1].

5. Terminology borrowed from [Ruffinelli et al., 2020].

6. Positive training examples are excluded from the development loss calculation to encourage better link predictions rather than reproducing the known facts.

7. ConvE embeddings are reshaped to (8,8) and (16,8) prior to convolution.

full). Similarly, the TransE (128 dimensions) model performs very well in terms of MRR but is the worst performing model with a global threshold and the second weakest (after ComplEx) in the KBC setting with multiple thresholds. The comparison of the model within the same type reveals, that models with a higher number of embedding dimensions perform better in MRR than models with a lower dimensionality, whereas model with less dimenstions are mostly better in the classification setting with a global threshold. The DistMult model pair constitutes an exception with DistMult (64) performing the best in all settings, however, the gap in MRR performance is significantly bigger than that in $F_1$ score.

We also provide an analysis in terms of the sets $\mathcal{C}$ (queries with exhaustive answers), $\mathcal{F}$ (queries with type violations), and $\mathcal{I}$ (queries with at least one removed answer due to entity removal). $\mathcal{C}$ corresponds roughly to the traditional setting, directly turned into a classification problem. Here, the queries always contain at least one answer and the amount of positives is unrealistically high. Adding[8] queries with type violations, i.e., $\mathcal{F}$, makes the problem more challenging and thresholding is more important in order to detect cases where no answer should be returned. However, the queries in $\mathcal{F}$ might still be too easy for approaches that are good at type modeling. $\mathcal{I}$ contains the most realistic (and most difficult to judge) empty queries and provides the most challenging scenario.

The full query set (full $= \mathcal{C} \cup \mathcal{F} \cup \mathcal{I}$) combines moderately challenging and hard cases, and measures progress at type modeling while at the same time containing realistic empty queries. For characterizing models in a nuanced evaluation, we suggest to report metrics for all of these three settings: full, $\mathcal{C} \cup \mathcal{F}$, and $\mathcal{I}$.

Table 2 shows that $\mathcal{C}$ queries always have the highest score, which makes sense as they can be considered the easiest setting. $\mathcal{C} \cup \mathcal{F}$ queries cause a strong drop in model performances compared to $\mathcal{C}$ when a global threshold is used but this difference largely disappears with threshold tuning. ConvE and ComplEx however still show a considerable drop for $\mathcal{C} \cup \mathcal{F}$ queries, which may indicate potential room for improvements through better type modeling. The $\mathcal{I}$ query set, in average, produces the lowest scores (except for ComplEx global threshold), which is in line with the intuition that this setting can be considered more difficult than $\mathcal{C} \cup \mathcal{F}$. TransE shows the least difference between $\mathcal{I}$ and $\mathcal{C} \cup \mathcal{F}$ (and $\mathcal{C}$), and has the highest score for $\mathcal{I}$ in both threshold settings — a remarkably stable performance across query sets.

**Qualitative analysis.** Table 3 sheds light on the exact model behavior in the two threshold settings by showing predictions for three sample queries from different sets. The highest-scored answers are generally thematically related to the correct answers. Relation-specific thresholds can improve the classification performance by including correct answers missed in the global setting (as is the case for the first query) but the opposite is also possible (second query). For the type-violating query[9], highest scores are obtained by entities with a semantic connection to the relation. ComplEx exploits the extreme threshold value of 1.0 to entirely disallow predictions for this relation.

---

8. Since $\mathcal{F}$ only contains queries without answers, it needs to be combined with, e.g., $\mathcal{C}$ for a meaningful computation of $F_1$.

9. The relation labels are simplified for readability. The original relation for this query is `/location/statistical_region/gni_per_capita_in_ppp_dollars`. `/measurement_unit/dated_money_value/currency`, which expects a location as its head.

The evaluation of these three queries also highlights the difference in ranking-based vs. classification-based evaluation approaches: despite the fact that ConvE ranks the candidates in a perfect order, the classification results still suffer from false positives and false negatives.

## 6 The Region model

To highlight the development potential of the existing models with respect to the new evaluation, we introduce a TransE variant which supports thresholds intrinsically. Relation-specific regions in the translation vector space — defined by up to $d * |\mathcal{R}|$ extra parameters — allow the model to better separate positive and negative predictions.

### 6.1 Definition

The distance function of *Region* maps to $\mathbb{R}_{0+}$, with 0 being the score for a perfect triple:

$$\delta(h, r, t) = (\mathbf{e_h} + \mathbf{r_r} - \mathbf{e_t})^T \mathbf{A_r}(\mathbf{e_h} + \mathbf{r_r} - \mathbf{e_t})$$

where $\mathbf{A_r}$ is a relation-specific positive semi-definite matrix that (together with a threshold) describes an elliptic region. A vector will be classified as positive if and only it is located inside this region. In order to limit the number of additional parameters in $\mathbf{A_r}$, we restrict $\mathbf{A_r}$ to be diagonal: $\mathbf{A_r} = diag(\mathbf{a_r})$ (allowing only positive values to ensure positive semi-definiteness).[10]

With a diagonal matrix $\mathbf{A_r}$, the computation can be simplified to (operand $\odot$ stands for element-wise multiplication, $\sqrt{\mathbf{x}}$ for element-wise square root):

$$\delta(h, r, t) = \|\sqrt{\mathbf{a_r}} \odot (\mathbf{e_h} + \mathbf{r_r} - \mathbf{e_t})\|_2^2$$

The transformation mentioned above is applied to the Region distances in the same manner to obtain a probability score:

$$s(h, r, t) = 1 - tanh(\delta(h, r, t))$$

TransE with L2 scoring is therefore a special case of Region with all $\mathbf{a_r}$ weights set to 1. Specifically, the original TransE model can be seen as a Region model with a fixed region radius shared across all relations.

### 6.2 Performance

Table 4 provides the evaluation results for the enriched model. The Region model achieves a noticeable improvement in terms of MRR and $F_1$ compared to TransE. While the Region model also improves in terms of the ranking metric MRR (16.5% relative increase), improvements for classification are particularly strong (32.4% and 36.8% relative improvements in terms of $F_1$ scores). With respect to different query sets, the biggest improvement is achieved in the complete and fake queries.

---

10. We also experimented with a spherical $\mathbf{a_r} \in \{a_r\}^d$, which gave slightly worse results.

| Query & Gold | global threshold | | multiple thresholds | |
|---|---|---|---|---|
| | ComplEx 128 | ConvE 128 | ComplEx 128 | ConvE 128 |
| Thomas Lennon – has profession – ? 

 ∈ *I* 

 **comedian** 
 **screenwriter** 
 **film producer** | *actor* 
 — th = 0.7 — 
 *tv producer* 
 *film director* 
 **screenwriter** 
 **film producer** 
 **comedian** 
 writer 
 musician | *actor* 
 **screenwriter** 
 **film producer** 
 *tv producer* 
 *film director* 
 — th = 0.5 — 
 **comedian** 
 musician 
 writer 
 author 
 artist | *actor* 
 *tv producer* 
 *film director* 
 **screenwriter** 
 **film producer** 
 — th = 0.1 — 
 **comedian** 
 writer 
 musician | *actor* 
 **screenwriter** 
 **film producer** 
 *tv producer* 
 *film director* 
 **comedian** 
 musician 
 — th = 0.3 — 
 writer 
 author 
 artist |
| Marion County – in time zones – ? 

 ∈ *C* 

 **Pacific** | — th = 0.7 — 
 Eastern 
 **Pacific** 
 Central 
 Mountain | **Pacific** 
 Eastern 
 — th = 0.5 — 
 Mountain 
 Central 
 Eastern European time in China 
 East Africa | Eastern 
 — th = 0.1 — 
 **Pacific** 
 Central 
 Mountain | **Pacific** 
 Eastern 
 Mountain 
 Central 
 — th = 0.3 — 
 Eastern European time in China 
 East Africa |
| ? – has currency – Meryl Streep 

 ∈ *F* | US dollar 
 euro 
 — th = 0.7 — 
 pound sterling 
 UK 
 Sweden | USA 
 UK 
 France 
 Ireland 
 Spain 
 Canada 
 — th = 0.5 — 
 Italy 
 New Zealand 
 Denmark | — th = 1.0 — 
 US dollar 
 euro 
 pound sterling 
 UK 
 Sweden | USA 
 UK 
 — th = 0.9 — 
 France 
 Ireland 
 Spain 
 Canada 
 Italy 
 New Zealand 
 Denmark |

Table 3: Exemplary predictions of Complex 128 and ConvE 128 models on the test queries with global and multiple thresholds. The first column presents the query, the corresponding query set and the gold answers in bold. Further columns show the top scored entities in the order of decreasing scores. Threshold values separate positive predictions (above the line) from negative predictions (below the line). Gold answers are also marked bold, correct answers contained in the train set are grayed out and italic (these answers are excluded from metric computation).

| Model (k) | MRR | F₁ global threshold | | | | F₁ multiple thresholds | | | |
|---|---|---|---|---|---|---|---|---|---|
| | | full | $\mathcal{C}$ | $\mathcal{C} \cup \mathcal{F}$ | $\mathcal{I}$ | full | $\mathcal{C}$ | $\mathcal{C} \cup \mathcal{F}$ | $\mathcal{I}$ |
| TransE 64 | .283 | .111 | .164 | .112 | .110 | .161 | .176 | .175 | .154 |
| Region ellipse 64 | **.330** | **.146** | **.266** | **.207** | **.123** | **.220** | **.317** | **.311** | **.162** |

Table 4: Performance of the Region model and the original TransE on the FB14k-QAQ test set.

## 7 Conclusion

This work points out the insufficiency of the current ranking-based evaluation paradigm for knowledge base completion and provides an alternative that directly measures the quality of predicted facts. We describe a process for constructing test collections that can measure KBC prediction quality and evaluate established KBC models on the new, carefully constructed FB14k-QAQ data set. Our experiments provide evidence that ranking-based estimation can be a misleading evaluation criterion for the actual completion task. With a simple but effective extension to the traditional TransE model, we encourage the research community to reconsider existing models in light of our more realistic evaluation setting and to conduct further research on factors that are crucial for classification performance. The new setup also allows to examine models with respect to how consistently scores are scaled across relationships and it motivates research on more universal and robust embedding models that reduce the performance gap between the global and multiple threshold settings.

## Acknowledgments

This work was funded by the Deutsche Forschungsgemeinschaft (DFG, German Research Foundation) - RO 5127/2-1, and by the BMBF as part of the project MLWin (01IS18050). We also thank our anonymous reviewers for their comments.

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

# Appendix A

**A.1**  A formal description of the query-based data set construction.

The ∘ operator is defined to fill the open position of a query with the specified entity, i.e., $q1 \circ i = (h, r, i)$ and $q2 \circ i = (i, r, t)$. For every *query*, a set of answers $\mathcal{A}_q^F$ can be extracted from a given set of facts $F \subseteq \mathcal{E} \times \mathcal{R} \times \mathcal{E}$, that contains the completions to valid facts, i.e. $\mathcal{A}_q^F = \{i \mid (q \circ i) \in F\}$.

Specifically, the following steps were taken during the construction of FB14k-QAQ:

1. Let $KB \subset \mathcal{E} \times \mathcal{R} \times \mathcal{E}$ be the underlying data set.

2. Let $Train \cup Valid \cup Test = KB$, $Train \cap Valid \cap Test = \emptyset$ the original partition of the $KB$. Unify the development and test data into single held-out set $H = Valid \cup Test$.

3. Randomly select a subset of entities $\mathcal{E}^- \subset \mathcal{E}$ that are to be removed. The rest of the entities $\mathcal{E}^+ = \mathcal{E} \setminus \mathcal{E}^-$ are the basis for the new data set.

4. Drop the triples from *Train* and $H$ where both head and tail entity were selected for removal, as they do not have any valid entities and do not suffice either for training or for query construction purposes:

$$Train \leftarrow Train \setminus \{(h, r, t) \in Train \mid h, t \in \mathcal{E}^-\}$$
$$H \leftarrow H \setminus \{(h, r, t) \in H \mid h, t \in \mathcal{E}^-\}$$

5. Move the triples with either one position selected for removal from *Train* to $H$ to be used for query construction, since the training process does not change and is still based on full triples.

$$temp \leftarrow \{(h, r, t) \in Train \mid h \in \mathcal{E}^-\} \cup \{(h, r, t) \in Train \mid t \in \mathcal{E}^-\}$$
$$H \leftarrow H \cup temp$$
$$Train \leftarrow Train \setminus temp$$

6. Transform the held-out set $H$ from triple form to query form. First, obtain the set of *answerable queries* that only include entities $\mathcal{E}^+$ and for which answers are contained in $H$:

$$\mathcal{Q} \leftarrow \{(h, r, ?) \mid h \in \mathcal{E}^+, \exists t : (h, r, t) \in H\} \cup$$
$$\{(?, r, t) \mid t \in \mathcal{E}^+, \exists h : (h, r, t) \in H\}$$

Second, extract the answer sets $A_q^H$ from the held-out triples $H$ for every query $\forall q \in \mathcal{Q}$. Note, that since the answers are retrieved from the held-out set only, and $H \cap Train = \emptyset$, these answer sets do not include any entities that complete a query to a triple from the train set, i.e. $\{q \circ i \mid i \in A_q \land q \in \mathcal{Q}\} \cap Train = \emptyset$.

7. Finally, the selected entities $\mathcal{E}^-$ are removed from the answer sets as well (the source of the answers $H$ is further omitted for readability reasons):

$$\forall q \in \mathcal{Q} : A_q \leftarrow A_q^H \setminus \mathcal{E}^-$$

Starting from this point, there are two types of queries regarding the completeness of their answer sets in the evaluation data of FB14k-QAQ:

Queries with exhaustive answers (complete): $\mathcal{C} = \{q \in \mathcal{Q} \mid A_q = A_q^H\}$

Queries with at least one removed answer (incomplete): $\mathcal{I} = \{q \in \mathcal{Q} \mid A_q \neq A_q^H\}$

Specifically, queries from $\mathcal{C}$ always contain at least one entity in their answer set, while queries from $\mathcal{I}$ can have empty answer sets. We will refer to these empty queries with no answer as $\mathcal{N} = \{q \in \mathcal{I} \mid |A_q| = 0\}$.

# Appendix B

**B.1** Threshold tuning algorithm. We used two tuning iterations over the relations (N=2).

---

**for** $r \in \mathcal{R}$ **do**:
    $\tau_r \leftarrow 0.5$
**end for**
$f1 \leftarrow 0.0$
$i \leftarrow 0$
**while** $i < N$ **do**:
    **for** $r \in \mathcal{R}$ **do**:                   ▷ in order of decreasing frequency of $r$ in the dev set
        **for** $\tau \in \{0.0, 0.1, 0.3, 0.5, 0.7, 0.9, 1\}$ **do**:
            $\hat{f}1 \leftarrow evaluate(r, \tau)$
            **if** $\hat{f}1 > f1$ **then**:
                $f1 \leftarrow \hat{f}1$
                $\tau_r \leftarrow \tau$
            **end if**
        **end for**
    **end for**
    i ← i+1
**end while**

---

**B.2** Tuned threshold statistics. The exact value is presented for the global threshold, aggregate statistics for multiple thresholds.

| Model ($d$) | global threshold | multiple thresholds | | |
|---|---|---|---|---|
| | | mean | min | max |
| ConvE 128 | 0.5 | 0.81 | 0.1 | 1 |
| ConvE 64 | 0.5 | 0.56 | 0.1 | 1 |
| ComplEx 128 | 0.7 | 0.63 | 0.1 | 1 |
| ComplEx 64 | 0.5 | 0.72 | 0.1 | 1 |
| TransE 128 | 0.1 | 0.26 | 0.1 | 0.9 |
| TransE 64 | 0.1 | 0.23 | 0.1 | 0.9 |
| DistMult 64 | 0.4 | 0.38 | 0.1 | 1 |
| DistMult 128 | 0.3 | 0.24 | 0.1 | 1 |
| Region 64 | 0.2 | 0.36 | 0.1 | 1 |

## Appendix C

**C.1**  Comparison of model performances on mean reciprocal rank and the $F_1$ score for the FB14k test (**dev**) set. The "$F_1$ global threshold" refers to scores obtained with a single shared threshold for all relations, while "$F_1$ multiple thresholds" refers to a setting with independent thresholds for every relation (including the inverse ones).

| Model ($d$) | MRR | $F_1$ global threshold | $F_1$ multiple thresholds |
|---|---|---|---|
| ConvE 128 | .3212 (.3279) | .134 (.135) | **.204** (**.218**) |
| ConvE 64 | .2633 (.2717) | .157 (**.155**) | .189 (.215) |
| ComplEx 128 | .2931 (.3009) | .021 (.022) | .190 (.200) |
| ComplEx 64 | .2931 (.3007) | .009 (.010) | .181 (.188) |
| TransE 128 | .2931 (.2988) | .108 (.103) | .159 (.160) |
| TransE 64 | .2834 (.2902) | .111 (.106) | .161 (.159) |
| DistMult 64 | .2663 (.2732) | **.159** (.153) | .184 (.210) |
| DistMult 128 | .2214 (.2310) | .133 (.146) | .163 (.189) |

**C.2:**  Overall performance records of precision, recall and $F_1$ (s. next page).

| Model ($d$) | | global threshold | | | | multiple thresholds | | | |
|---|---|---|---|---|---|---|---|---|---|
| | | $\mathcal{C}$ | $\mathcal{C} \cup \mathcal{F}$ | $\mathcal{I}$ | full | $\mathcal{C}$ | $\mathcal{C} \cup \mathcal{F}$ | $\mathcal{I}$ | full |
| ConvE 128 | prec | .232 (.228) | .155 (.160) | .061 (.063) | .083 (.084) | .312 (.317) | .258 (.269) | .116 (.129) | .167 (.176) |
| | rec | .329 (.325) | .329 (.325) | .370 (.371) | .351 (.351) | .322 (.331) | .322 (.331) | .211 (.253) | .261 (.288) |
| | $F_1$ | .272 (.268) | .211 (.214) | .105 (.107) | .134 (.135) | .317 (.317) | .286 (.297) | .150 (.171) | .204 (.218) |
| ConvE 64 | prec | .399 (.396) | .274 (.264) | .076 (.078) | .123 (.119) | .370 (.371) | .291 (.311) | .107 (.140) | .165 (.193) |
| | rec | .250 (.245) | .250 (.245) | .184 (.203) | .214 (.222) | .270 (.276) | .270 (.276) | .182 (.219) | .222 (.244) |
| | $F_1$ | .307 (.303) | .261 (.254) | .108 (.113) | .157 (.155) | .312 (.316) | .280 (.292) | .135 (.170) | .189 (.215) |
| ComplEx 128 | prec | .384 (.370) | .009 (.010) | .038 (.034) | .012 (.013) | .282 (.284) | .225 (.242) | .107 (.113) | .150 (.154) |
| | rec | .108 (.104) | .108 (.104) | .048 (.040) | .075 (.068) | .311 (.323) | .311 (.323) | .215 (.256) | .259 (.286) |
| | $F_1$ | .169 (.163) | .017 (.019) | .042 (.037) | .021 (.022) | .296 (.302) | .261 (.277) | .143 (.157) | .190 (.200) |
| ComplEx 64 | prec | .202 (.200) | .002 (.002) | .037 (.038) | .005 (.005) | .298 (.296) | .227 (.248) | .105 (.109) | .143 (.151) |
| | rec | .128 (.126) | .128 (.126) | .124 (.128) | .126 (.127) | .275 (.267) | .267 (.275) | .224 (.227) | .244 (.249) |
| | $F_1$ | .157 (.155) | .005 (.005) | .057 (.058) | .009 (.010) | .282 (.285) | .245 (.261) | .143 (.148) | .181 (.188) |
| TransE 128 | prec | .140 (.146) | .075 (.087) | .063 (.057) | .066 (.064) | .199 (.200) | .190 (.193) | .106 (.114) | .123 (.132) |
| | rec | .181 (.183) | .181 (.183) | .407 (.334) | .304 (.267) | .151 (.156) | .151 (.156) | .282 (.241) | .222 (.203) |
| | $F_1$ | .158 (.162) | .106 (.118) | .108 (.098) | .108 (.103) | .172 (.175) | .168 (.172) | .154 (.154) | .158 (.160) |
| TransE 64 | prec | .143 (.146) | .079 (.088) | .063 (.059) | .067 (.066) | .195 (.196) | .194 (.195) | .106 (.112) | .124 (.131) |
| | rec | .192 (.191) | .192 (.191) | .424 (.340) | .319 (.274) | .160 (.159) | .160 (.159) | .282 (.235) | .226 (.201) |
| | $F_1$ | .164 (.166) | .112 (.120) | .110 (.100) | .111 (.106) | .176 (.175) | .175 (.175) | .154 (.151) | .160 (.158) |
| DistMult 64 | prec | .362 (.389) | .233 (.266) | .089 (.082) | .121 (.117) | .254 (.377) | .222 (.320) | .116 (.133) | .153 (.188) |
| | rec | .219 (.214) | .219 (.214) | .238 (.219) | .229 (.217) | .258 (.268) | .258 (.268) | .204 (.211) | .229 (.237) |
| | $F_1$ | .273 (.276) | .226 (.237) | .129 (.119) | .159 (.152) | .256 (.314) | .239 (.292) | .148 (.163) | .184 (.208) |
| DistMult 128 | prec | .317 (.410) | .165 (.263) | .076 (.075) | .116 (.140) | .181 (.376) | .163 (.302) | .113 (.129) | .135 (.195) |
| | rec | .218 (.223) | .218 (.223) | .105 (.095) | .156 (.152) | .239 (.246) | .239 (.246) | .179 (.134) | .206 (.184) |
| | $F_1$ | .275 (.289) | .188 (.242) | .088 (.083) | .133 (.146) | .206 (.298) | .194 (.271) | .138 (.131) | .163 (.189) |

Table 5: Precision, recall and $F_1$ scores on different query sets for the FB14k test set. "global threshold" refers to scores obtained with a single shared threshold for all relations, while "multiple thresholds" refers to a setting with independent thresholds for every relation. (Dev set scores in parentheses.)