# OpenReview forum: "Ranking vs. Classifying: Measuring Knowledge Base Completion Quality"
_AKBC.ws/2020/Conference — AKBC 2020_

### Official Review · AnonReviewer2 · 2020-03-09
**KBC evaluation paper, more experiments needed**

**Rating:** 3
**Confidence:** 5

**Review:**

Summary: The paper proposes a new evaluation paradigm for KB completion methods that focuses on classificiation, rather than ranking. The authors construct an alternative dataset FB13kQAQ which contains false as well as true facts. They analyse the performance of existing KBC methods (DistMult, ComplEx, etc.) on the new dataset and propose a new KBC model, which can be seen as a variant of TransE with thresholding.

Whilst I agree with the general premise of developing a better way for evaluating KBC methods, I believe the paper in its current state is not ready for publication. More details below:

Section 3:
1. I'm not sure how useful type violation queries are, as those shouldn't be particularly hard to predict. Wouldn't it be better to look at ranked predictions of existing state-of-the-art models and extract incorrect queries that are ranked highly to create a hard dataset for future KBC work?
2. Writing quality should be improved. In particular, the dataset creation process should be described in a clearer and much more succint way.

Section 4:
1. No ned to describe Algorithm 1 in such detail on almost half a page. This space could be used for additional experiments (see below).

Section 5:
1. Since BCE is used as a loss function and a logistic sigmoid is applied to every triple, this gives a natural classification threshold of 0.
	a) It would be interesting to see how this simple baseline threshold compares to the tuned one.
	b) Additionally, one could add a relation-specific bias to every scoring function that is then learned, rather than using a separate tuning strategy for the relation-specific thresholds.
2. It would be nice to see models compared with the same number of parameters, as opposed to the same embedding dimensionality (e.g. ComplEx has 2x as many parameters as DistMult for the same embedding dimensionality).
3. The authors don't provide an analsys of model performances on S, M, N and F queries separately.
4. Some qualitive analysis on what kind of queries particular models struggle with would be nice to see.

Section 6:
1. The authors don't provide a clear motivation for the newly proposed model and why it should be better than the existing models. Why does A_r have to be a diagonal positive semi-definite matrix? In general, the new model does not seem like a natural add-on to a paper focused on KBC evaluation and should perhaps be separated into a different paper and replaced by more extensive experiments (see above).
2. The proposed Region model outperforms ConvE on the F1 score and performs comparably on the MRR. Since ConvE is a relatively old model at this point, the Region model should however be compared to more recent state-of-the-art models, e.g. RotatE (Sun et al., ICLR 2019.) and TuckER (Balazevic et al., EMNLP 2019).
3. How does the proposed model compare on the existing WN18RR and FB15k-237 datasets?

Overall, the abstract should be made much more succint and the writing quality of the whole paper could be improved.

---

> ### Author Response · Authors · 2020-04-15
> **Response to  AnonReviewer2's review**
>
> We thank the reviewer for the thoughtful comments.
>
> General note:
>
> Due to a recently discovered bug, the revised paper reports slightly different numbers than the original submission. The conclusions are still valid.
>
> Responses to the questions raised:
>
> *Usefulness of type violations*
> => We added additional analysis that shows that while queries based on type violations are easier for models than queries constructed by entity removal, type violations still constitute a significant challenge to all models considered. We believe they cover an aspect that is important to measure.
>
> *Use predictions of model for test set creation*
> => We decided to use an approach that is independent of already existing models for test set construction, since we wanted to avoid any biases in the test set that would reflect biases of specific models.
> We believe that only relying on properties of the underlying data set makes the evaluation setting more useful in the long term.
> Moreover, given that most models still struggle to reach a performance > 20%, our primary intention was not to create a particularly *hard* evaluation setup, but one that realistically measures performance in a KB Completion (i.e., KB entry prediction) setting.
>
> *Clarify data set description*
> => We shortened the description of the data set construction and provided an additional diagram to increase readability.
>
> *No need to describe Algorithm 1*
> => Moved to appendix.
>
> *BCE with threshold 0.5*
> => The global threshold experiments include runs with a threshold of 0.5.
> This classification threshold would however only be natural if the evaluation setting mirrored the training setting, which is not the case for KBC.
> It is therefore natural to find a threshold that is adapted to the evaluation setting of interest (finding new facts, rather than separating positive from negative training samples).
> We included insight over the global thresholds used in Appendix B.2.
>
> *relation-specific bias*
> => Similarly, relation specific biases would add more capacity to the model to capture positive/negative ratios of sampled facts during training.
> Whether such an additional modeling capacity is beneficial (i.e. generalizes to the training data), or leads to overfitting on the training data, is an interesting question that could be examined with the evaluation setup that we propose.
> In the current work however, we decided to use threshold tuning instead, which should have an effect similar to including a bias in the model.
>
> *number of parameters of models*
> Rather than trying to establish comparisons where choices of hyper-parameters (e.g. number of parameters) are exactly the same between models, we performed a limited hyper-parameter search (across values reported as standard for this task) and compared the results.
> The point of the comparison is to highlight that the architectures behave differently in the new evaluation setting.
>
> *Analysis of different query types*
> => We included a detailed analysis for different query types.
>
> *Qualitative analysis*
> => We included a qualitative analysis showing actual queries and answers found by different models.
>
> *Newly proposed model*
> The motivation for the newly proposed model is that A_r explicitly describes a decision boundary for positive/negative by an ellipse (which requires positive semi-definiteness).
> Otherwise, the underlying mechanism is the same translational model as in TransE.
> Our intention was to show that a simple additional component (modeling a decision boundary by an ellipse modulo a threshold) can improve performance in a classification scenario.
> We agree that the newly proposed model should not be viewed as the main contribution of the paper, and we shortened Section 6.
>
> *Shorten abstract*
> We shortened the abstract to be more succinct.

---

> > ### Comment · AnonReviewer2 · 2020-04-16
> > **Response after rebuttal**
> >
> > I would like to thank the authors for addressing some of my concerns. Overall, I believe the paper has improved considerably compared to the initial submission. However, I still have some major concerns due to which I stand by my decision that the paper is not acceptable in its current state:
> >
> > 1) The performance of ComplEx for global thresholds in Table 1. seems suspiciosly low compared to DistMult, given the similarity between those two models. I would expect the two models to perform similarly.
> >
> > 2) Comparison with some more up-to-date SOTA models, such as RotatE (Sun et al., ICLR 2019.) and TuckER (Balazevic et al., EMNLP 2019) is still missing. These models have been shown to improve on TransE when comparing MRRs by a great margin. It would be important to see how the proposed improvement on TransE compares against these more recent models on both MRR and the proposed classification setting.

---

> > > ### Author Response · Authors · 2020-04-22
> > > **Author response**
> > >
> > > We thank the reviewer for sharing their concerns.
> > >
> > > *ComplEx results*
> > >
> > > We checked again that the results for ComplEx (obtained, as for all other methods, with the framework of Dettmers et al., AAAI 2018) are, to the best of our knowledge, indeed valid.
> > >
> > > As an indicator of general correctness, consider the performance on MRR, where the results for ComplEx fit into the model ranking reported elsewhere (e.g. Trouillon et al., ICML 2016; Dettmers et al., AAAI 2018) and are better than the results for DistMult. It might seem counter-intuitive that these models perform differently on the global threshold setting, however, it highlights the differences in per-relation scaling (that disappears for MRR and per-relation thresholding as expected).
> > >
> > > Further investigation of tuned threshold values showed that across all models, the tuned thresholds of ComplEx models have the highest variance (i.e. they deviate most from the mean threshold value). This indicates a strong calibration mismatch among scores for different relations, which makes the global threshold setting much more challenging for some models.
> > >
> > > Thereby the introduced evaluation reveals potential for further hyper-parameter tuning, regularization adjustments, etc.
> > >
> > >
> > > *Comparison to more recent approaches*
> > >
> > > We agree that an evaluation on a broader set of models including TuckEr (Balazevic et al., EMNLP 2019) would provide a richer overview, but unfortunately we had to limit the model selection due to time restrictions. We would like to emphasize that our smaller model set contains models from different approach groups, which we consider sufficient for the show-case.  Furthermore, RotatE results are reported to be comparable to ConvE when used with the same negative sampling scheme (Balazevic et al., EMNLP 2019).
> > >
> > > Regarding the TransE improvement, it should be considered independently from the SOTA models and focuses on a relative improvement to TransE.

---

### Official Review · AnonReviewer1 · 2020-03-27
**The authors convincingly argue that ranking-based metrics (like MRR) for entity-relation completion are misleading for many downstream tasks.**

**Rating:** 7
**Confidence:** 4

**Review:**

The main contributions of the paper are a new train/test dataset that provides negative cases and queries to fulfill. Based on this dataset the authors show how to adapt a traditional IR methodology with F1, which is compared to current measurement techniques, especially MMR. Finally, the authors provide a variant of TransE that yields significant improvement in F-Score.

The paper is overall interesting, well-written and easy to follow, modulo a few specific points (see below).

The idea of introducing new metrics for entity-relation completion tasks that are closer to real-life downstream tasks is compelling and clearly conveyed. Good MRR rankings were useful for entity-relation predictions initially, but as quality rises, better metrics are necessary.

The result section shows how to apply the new metric compared to standard metrics, which is convincing.


Detailed comments:

The exact process used to build the new dataset is unclear. To clarify it, it would be beneficial to report the numbers after every step. Also, it would be great to add an example for step 6 to better understand what "answerable queries"  refers to exactly, as well as one example per query category (multiple answers, single answer, no answer).

Also, it would be interesting to measure how the performance evolves with the overall size of the graph (in terms of the various measurement methods).

Finally, it would be good to have a more detailed discussion regarding how the different measurement metrics differ. The text mentions that there is "almost no correlation"; would it be possible to give substance to this claim with some numbers and to explain this further?

If space is an issue to add those various points, the authors could remove or shorten the simple variant of TransE.

---

> ### Author Response · Authors · 2020-04-15
> **Response to AnonReviewer1's review**
>
> We thank the reviewer for the thoughtful comments.
>
> General note:
>
> Due to a recently discovered bug, the revised paper reports slightly different numbers than the original submission. The conclusions are still valid.
>
> Responses to the questions raised:
>
> *Clarify data set description*
> => We visualized description of the data set construction in a corresponding diagram to increase readability.
>
> *Report numbers for query sets.*
> =>  We added more detailed numbers/set sizes for central steps of the data set creation.
> In particular, we added the number of entities selected for removal, and the sizes for the sets C, I, and F (see appendix A.2).
>
> *Examples, examples for answerable queries*
> => We show example queries in Section 5.
> For answerable queries answers are shown in bold.
>
> *measure performance w.r.t. size of the graph*
> => While it would be a possible evaluation aspect, within the scope of our paper we tried to focus on the obtained split and did not experiment with different graph sizes.
>
> *more detailed discussion*
> => We expanded the analysis and discussion.
> (We reformulated the statement "almost no correlation". What we wanted to express is simply that the rankings of methods by metric in Table 1 are not the same for different metrics.)
>
> *shorten the simple variant of TransE*
> We shortened Section 6.

---

### Official Review · AnonReviewer4 · 2020-03-31
**Interesting new evaluation, could use more analysis**

**Rating:** 6
**Confidence:** 4

**Review:**

Summary:
The authors propose a new classification-based evaluation approach for knowledge base completion models. They create new dev/test sets by adding negative examples created by a combination of filtering out entities from true facts and creating type-inconsistent entity-relation pairs. They then evaluate a handful of state-of-the-art embedding models using this new metric. Lastly, they propose a new embedding scoring function for TransE that improves performance on the classification evaluation over the original TransE.

Clarity:
The explanations of the dataset and model are clear, but could be shorter. The motivation/how it differs from prior work (like the classification evaluation used in Socher 2013) could use more explanation, or perhaps concrete examples.

Originality and Significance:
The evaluation metric and modification to TransE are both novel. This approach to classification seems like it would give a better indication of model quality than classifying (possibly) perturbed triples as has been done in the past.

Pros:
- Useful evaluation metric for KBC
- Methodology is clear

Cons:
- Limited analysis of the evaluation. Some more discussion/inspection of why certain models perform the way the do could help show what aspects the classification captures that MRR doesn't.

Other comments:
- I would be curious to see some breakdown of how models perform on the different subsets of the evaluation data. For
  example, I would expect the embedding models would generally do better on the type-inconsistent queries compared
  to the missing entities.

---

> ### Author Response · Authors · 2020-04-15
> **Response to AnonReviewer4's review**
>
> We thank the reviewer for the thoughtful comments.
>
> General note:
>
> Due to a recently discovered bug, the revised paper reports slightly different numbers than the original submission. The conclusions are still valid.
>
> Responses to the questions raised:
>
> *Shortening data set description*
> => We provided a simpler description of the data set construction with a corresponding diagram to increase readability, which may, however, include some ambiguities. We believe that details are important here (e.g., what happens if an answer to a query in the test set was originally in the train split?), and that a rigorous way of describing the process is the only way to avoid misunderstandings and gaps. Therefore a detailed description is still available in the Appendix A.1.
>
> *Motivation w.r.t. prior work*
> => We expanded the corresponding part of Section 2, related work.
>
> *More analysis of evaluation / evaluation on subsets*
> => We added additional quantitative analysis by splitting the test queries into those based on type violations, and those based on entity removal. We also added more qualitative analysis by inspecting concrete queries.

---

### Decision · Program_Chairs · 2020-04-30

**Decision:**

Accept

**Comment:**

This paper proposes a new evaluation for KBC where models need to decide whether to accept a new fact instead of simply ranking the possibilities. The main contribution of this work is the well-motivated evaluation that is better aligned with how these models would in practice be used downstream. There is a secondary contribution of a variant of TransE that is tailored towards the more realistic setting reflected by the evaluation. While there are concerns about the lack of more recent models,  the novel method serves to highlight the goal of the new evaluation rather than to claim state-of-the-art performance.